# Free Radical Copolymerization of Diallylamine and Itaconic Acid for the Synthesis of Chitosan Base Superabsorbent

**DOI:** 10.3390/polym14091707

**Published:** 2022-04-22

**Authors:** Wafa Al-Mughrabi, Abeer O. Al-dossary, Abir Abdel-Naby

**Affiliations:** 1Department of Chemistry, College of Science, Imam Abdulrahman Bin Faisal University, P.O. Box 1982, Dammam 31441, Saudi Arabia; walmagribi@iau.edu.sa (W.A.-M.); aodossary@iau.edu.sa (A.O.A.-d.); 2Water Treatment Unit, Basic & Applied Scientific Research Center (BASRC), Imam Abdulrahman Bin Faisal University, P.O. Box 1982, Dammam 31441, Saudi Arabia

**Keywords:** free radical polymerization, superabsorbent, water-retaining agent, thermal properties

## Abstract

Copolymerization of diallylamine (DAA) and itaconic acid (IA) was synthesized using benzoyl peroxide as a free radical initiator, in dioxane as the solvent. The composition of the copolymer was determined by the nitrogen content using Edx. The solubility of the copolymer was also investigated. The water solubility of the synthesized copolymer depends on the comonomers’ ratio. The structure of the copolymer was confirmed by ^13^C-NMR spectroscopy. To increase the water insolubility of the copolymers, and keep their hydrophilicity, the copolymer was allowed to react with chitosan to form a superabsorbent polymeric material (SP). The structure of the synthesized superabsorbent was confirmed using ^13^C-NMR spectroscopy. The thermal property of the (SP) was also investigated by TGA. The investigation of the chitosan-based superabsorbent, as water-retaining agents, was studied. The results revealed that the superabsorbent polymers exhibited a good swelling ability and salt tolerance.

## 1. Introduction

Water is considered an essential need for living creatures, humans, animals, and plants. It is also important for various activities such as agriculture, especially in countries exhibiting a desert nature [1,2,3]. Superabsorbent polymers (SA) are hydrophilic polymers, that can absorb and retain water for short time but do not dissolve in water [4,5,6]. The (SA) polymers are widely used in various applications, such as agriculture, industry, drug delivery, and personal care [7,8,9]. In agriculture, the (SA) was used to reduce the frequency of irrigation, as it improves soil water retaining which enhances the plant survival rates, especially in the desert [10,11,12,13,14]. As an environmental requirement, naturally based superabsorbents are always used, such as starch, cellulose, proteins, and chitosan [15,16,17,18].

Chitosan is one of the most abundant natural polymers. It is used in various applications because of its compatibility, degradability, and nontoxicity [19]. Despite these advantages, chitosan suffers from low thermal stability at high temperatures. Additionally, the high percentage of hydrogen bonding limits its adjustment to various applications without modification [20,21]. Chitosan was known to undergo graft copolymerization with vinyl monomers, such as acrylic acid, acrylamide, and acrylonitrile [22,23,24].

Diallylamine (DAA) is known to undergo copolymerization by free radical mechanism, forming pyrrolidine rings [25]. The pyrrolidine ring is considered a proton adsorption site. Itaconic acid exhibits two carboxylic groups and an ethene bond, which enable it to undergo both copolymerization and polycondensation. IA and its producing special hydrogels for water decontamination, targeted drug delivery as well as smart nanohydrogels in food applications, coatings, and elastomers [26].

In the present work, copolymerization of DAA and IA with chitosan will be synthetized to produce a superabsorbent chitosan base polymeric material for the performance of hydrogels.

## 2. Experimental Section

### 2.1. Materials

Chitosan (CS), degree of acetylation 80%, average molecular weight 50,000, was purchased from Sigma-Aldrich. Itaconic acid, sodium bisulfite, and benzoyl peroxide were obtained from Loba Chemie. Potassium persulphate was obtained from Winlab limited. Ethanol, acetic acid, and DAA were obtained from Sigma-Aldrich.

All chemical reagents are of analytical grade.

### 2.2. Copolymerization of Diallylamine and Itaconic Acid

Various in-feed concentrations of the two comonomers were allowed to undergo free-radical copolymerization, in dioxane, using 0.02 M benzoyl peroxide as the initiator. The reaction was carried out under a nitrogen atmosphere for certain intervals of time, at 60 °C, in an ultrasonic bath of power 300 watts. The copolymer was filtered with diethyl ether and then washed with ethanol using Soxhlet system to remove any homopolymer. The copolymer was dried, weighted, and the found composition of the copolymer was determined by deducing the nitrogen content of the prepared copolymer.

### 2.3. Synthesis of the Superabsorbent Polymers

Synthesis of the superabsorbent polymer was carried out in two-neck round-bottom flask. A pure (2 g) chitosan was dissolved in 100 mL of 1% acetic acid solution. Proper concentrations of initiators (0.01 M, 0.015 M, 0.02 M) of Sodium bisulfite and potassium persulfate were added at temperatures (30 °C, 40 °C, 50 °C, 60 °C), in ultrasonic bath of 300 watts, under nitrogen atmosphere. After 15 min the comonomers were successively added. The reaction was carried out for a given interval of time (1–6 h). The Soxhlet extraction system was used to remove the homopolymers. After drying, the graft copolymer was weighted and the graft percentage (*G%*) was calculated according to the following equation:$$G\%=\frac{w-w_{o}}{w_{o}}\times 100$$

*w_o_* = original weight of chitosan, *w* = weight of the graft copolymer.

### 2.4. Spectroscopic Analysis

The structure of the copolymer and the superabsorbent polymer was confirmed using solid-state NMR Bruker Avance III spectrometer, operating at 400 MHz.

### 2.5. SEM and Energy Dispersive Spectrometer (EDS)

A VEGA 3 TESCAN scanning electron microscope (Tescan, Czech Republic) with a detector of secondary electron (SE). In addition, detector and energy-dispersive spectrometer (EDS) were used to determine the nitrogen content of the copolymer formed. The analysis was carried out at voltage of 15 KeV with a working distance of 10 mm between the specimen and the detector.

### 2.6. Swelling Measurements

The superabsorbent powder (0.05 g) was dispersed in distilled water (500 mL) for 4 h, at room temperature to reach the swelling equilibrium. The residual water was removed by filtration using a 100 mesh stainless steel screen until water ceased to drip.

The water absorbency was calculated according to the following Equation (1)
(1)$$Q=\frac{\left(m_{1}-m_{0}\right)}{m_{0}}$$
where *Q* is the water absorbency (*w/w*), *m*_0_ is the weight of the dry superabsorbent polymer, and *m*_1_ is the swollen superabsorbent polymer.

### 2.7. Swelling in Salt Solutions 

The absorbency of the superabsorbent polymer was evaluated in variable NaCl solutions (from 0.1 to 1.0 *w/w*%), using the above method described for the swelling measurements in distilled water.

### 2.8. Water Retention of the Superabsorbent Polymers 

The water retention of the superabsorbent polymer (SA) was tested using the following method.

A specific amount of (SA) was allowed to swell to saturation in distilled water. The superabsorbent polymers were filtered using a 100 mesh screen and placed in Petri dishes, at room temperature. The weight of SA was determined after regular time intervals. The process was continued until saturation (no change in weight) was detected.

The water retention was obtained by applying the following equation: (2)$$water\;retention\left(\%\right)=\frac{\left(wt-wd\right)}{\left(wi-wd\right)}\times 100$$
where, *wt* is the weight of SA, at time (t), *wd* is the weight of the dry superabsorbent polymer, and *wi* is the initial weight of swollen superabsorbent polymer.

## 3. Results and Discussion

### 3.1. Copolymerization of DAA and IA

The composition of the copolymer was determined according to the nitrogen content, as the nitrogen atom is a direct confirmation of the DAA moieties present in the synthetized copolymer (Table 1).

The structure of the copolymer formed in sample 2 was confirmed using ^13^C-NMR (Figure 1).

The results revealed that the DAA cyclo-polymerized to form pyrrolidine rings [25], while the IA formed a condensation product with the pyrrolidine -NH group. Moreover, the presence of the ethylenic carbons accounted for the low composition of IA in the synthetized copolymers (Table 2).

### 3.2. Synthesis of the Superabsorbent Polymer 

Although IA comonomer exhibits two carboxylic groups, which enable it to absorb either water or salty water, it also increases its water solubility. To profit from the advantage of the presence of carboxylic groups and overcome the water solubility, graft copolymer of both comonomers with chitosan (CS) occurred using redox polymerization. 

#### 3.2.1. Effect of Various Parameters on the Percentage of Graft

The following equation was used to calculate the grafting percentage
(3)$$G=\frac{W-W_{0}}{W_{0}}\;\times \;100$$

To synthetize insoluble superabsorbent, the comonomers concentrations to the chitosan were kept (CS = 2 g, [DAA] = 0.4 M, [IA] = 0.15 M).

##### Effect of Time 

The effect of various intervals of time on the grafting copolymerization of comonomers onto CS is listed in Table 3. The results revealed that the percentage of the graft increased gradually with time. This might be attributed to the increase in the chain length of the grafting branches until it reached a maximum value. Afterward, the increase in the reaction time led to a steady state for the G%, due to the consumption of monomer units in the formation of branches. 

##### Effect of Temperature 

To investigate the effect of reaction temperature on the graft copolymerization reaction, the temperature was increased from room temperature to 60 °C. The maximum percentage of the graft was achieved at 40 °C. Afterward, a gradual decrease in the percentage of the graft was observed with the increase in temperature, which could be attributed to the achievement of ceiling temperature of the polymeric branches as shown in Table 4.

##### Effect of Initiators Concentration 

The effect of initiator concentration on the percentage of graft onto Chitosan is listed in Table 5. A gradual increase in G% was observed with the increase in initiator concentration from 0.015 M to 0.02 M. Afterward, a decrease in the G% was observed by increasing the initiator concentration, which could be ascribed to the increase in the probability of chain transfer to initiator reactions. 

Thus, the optimum conditions for the synthesis of the superabsorbent were (CS = 2 g, [DAA] = 0.4 M, [IA] = 0.15 M, [I] = 0.02 M, and T = 40 °C).

##### 3.2.2. ^13^C-NMR Spectroscopic Analyses

The CS-g-(DAA-IA) ^13^C-NMR spectrum confirmed the structure of the superabsorbent polymer (Figure 2). The results showed the condensation of the IA carboxylic group with an amino group of chitosan (
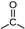
 = 174 ppm) as well as the condensation of the IA carboxylic group with -NH groups of pyrrolidine ring (
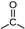
 = 175 ppm). In addition to another two types of carbonyl groups of IA moieties at 176.5 ppm and that of chitosan amino group at 178 ppm. The ethylenic bond was confirmed by the peaks at 129 ppm, 136 ppm.

From the above data, one can determine the structure of SA as follows:



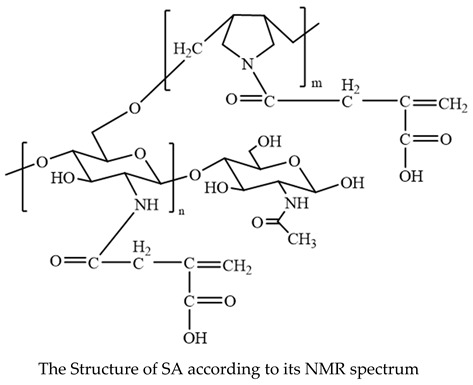



Another confirmation of the modification of chitosan is the surface morphology of the superabsorbent as compared to parent chitosan (Figure 3).

Figure 3b shows the branches built as extra layers on the top of the main chitosan chains. In addition, the chitosan shows smooth morphology, while the superabsorbent showed a rough surface due to the branches formed by the pyrrolidine moieties. The surface roughness is known to enhance the water permeability [27].

### 3.3. Thermal Properties of the Superabsorbent

To adjust the synthetized superabsorbent to any application, its thermal behavior should be examined. Figure 4 shows the TGA curves of various graft copolymers as compared to that of the native chitosan.

The results revealed that the graft copolymerization affected the initial decomposition temperature (T_o_), the temperature at which the polymer starts to lose part of its polymeric matrix. This is attributed to the decrease in the matrix crystallinity as a consequence of the formation of branches onto the chitosan main chains. Despite the decrease in T_o_ values, the thermal stability of the graft copolymers could be shown by the decrease in weight loss percentages at high temperatures as compared to the native chitosan, which lost almost half of its weight at 500 °C. Thus, the increase in the percentage of the graft gave the copolymer its extra thermal stability.

### 3.4. Water Absorbency Measurements

#### 3.4.1. Effect of the Percentage of Graft

The effect of comonomers contents on the water absorption of the superabsorbent is shown in Figure 5.

The results revealed that the increase in the graft percentage led to an increase in the water absorbency. This is attributed to the increase in the number of carboxylic groups present in the superabsorbent polymer as itaconic acid moieties.

#### 3.4.2. Swelling in Salt Solutions

The absorbency of the superabsorbent polymer in different salt concentration solutions is illustrated in Figure 6.

The superabsorbent polymer swelling in different salt concentration solutions is illustrated in Figure 6. The results revealed that the water absorption capacity decreased with the increase in salt concentration. This is attributed to a charge screening effect of additional cations, causing anion–anion electrostatic repulsion [27].

The absorbency of SA in the salt solution is higher than that of CS. This is attributed to the basicity of the pyrrolidine ring to attract the protons of carboxylic groups of IA and the chlorine anion of NaCl to form quaternary ammonium salt.

From the above-mentioned data, the swelling efficiency of the superabsorbent polymer depends on its chemical structure and the medium.

### 3.5. Water Retention

The water retention property of SA was investigated. The results are illustrated in Figure 7.

The results revealed that the water retention of SA decreased with time. The superabsorbent polymer (%G = 60.65) exhibited higher water retention efficiency as compared to CS, as it retained up to 50% of water after 40 h. This is attributed to the ability of the SA polymer to undergo hydrogen bonding with water molecules [28].

## 4. Conclusions

A novel superabsorbent copolymer was successfully synthesized by the free radical polymerization of chitosan with diallylamine and itaconic acid comonomers. The reaction conditions were optimized. The structure of the superabsorbent was confirmed by ^13^C NMR. The diallylamine was found to form polymeric branches as pyrrolidine rings while itaconic acid formed condensation products with the amino group of chitosan and the -NH group of pyrrolidine rings. The superabsorbent exhibited better thermal stability as compared to the native chitosan. The results of the effect of comonomers contents on the water absorption of the superabsorbent revealed that the increase in the graft percentage led to the increase in the water absorbency. This is attributed to the increase in the number of carboxylic groups present in the superabsorbent matrix as itaconic acid moieties. The absorbency of SA in the salt solution is higher than that of CS, this is attributed to the basicity of the pyrrolidine ring to attract the protons of carboxylic groups of IA moieties and the chlorine anions of NaCl salts to form quaternary ammonium salt.

The superabsorbent polymer (%G = 60.65) exhibited higher water retention efficiency, as compared to CS, as it retained up to 50% of water after 40 h. This is attributed to the ability of the SA polymer to undergo hydrogen bonding with water molecules.

The new superabsorbent is recommended to become an ideal soil water retention agent.

## Figures and Tables

**Figure 1 polymers-14-01707-f001:**
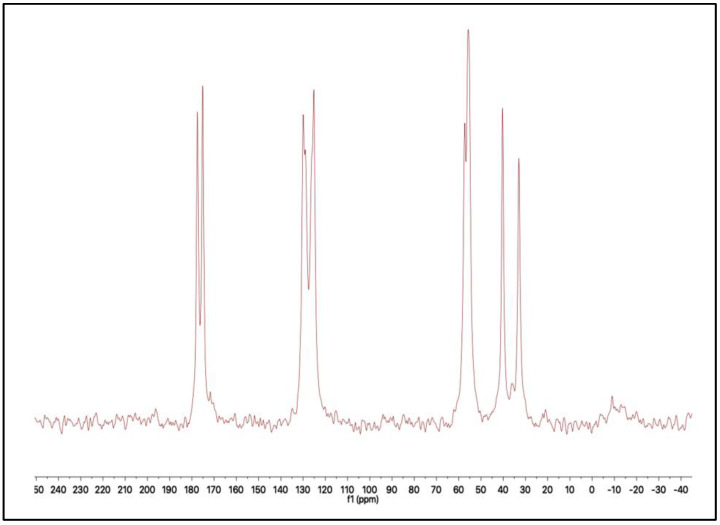
^13^C-NMR spectrum of (DAA:IA) copolymer (sample 2).

**Figure 2 polymers-14-01707-f002:**
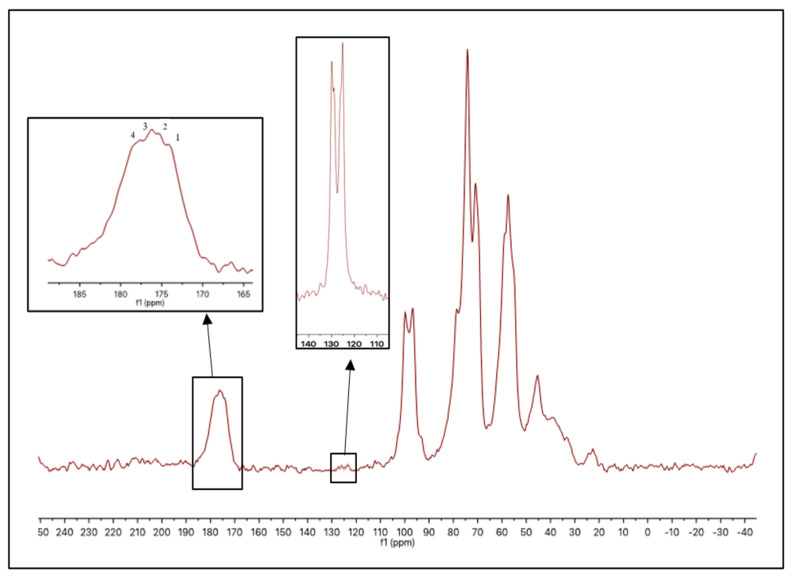
^13^C-NMR spectrum of SA.

**Figure 3 polymers-14-01707-f003:**
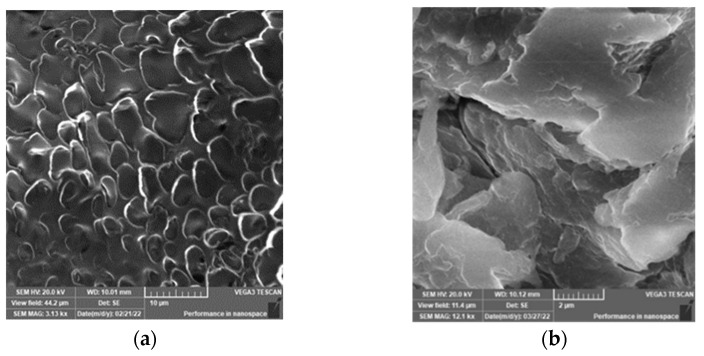
SEM Morphology of the superabsorbent (**b**) as compared to unmodified chitosan (**a**).

**Figure 4 polymers-14-01707-f004:**
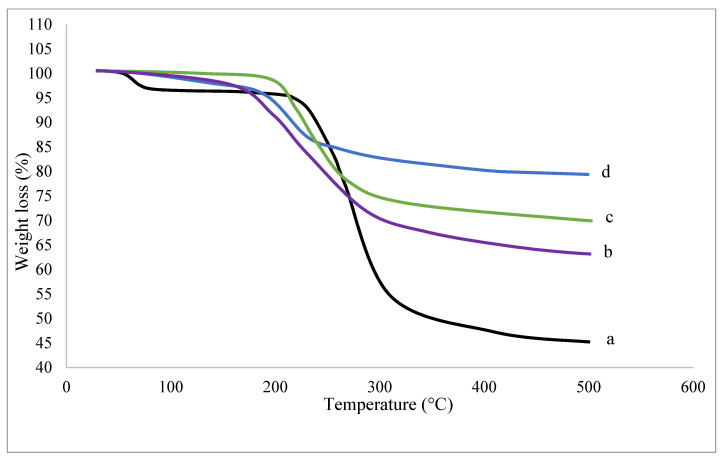
TGA of various CS-g-(DAA-IA) copolymers (%G: b = 49.1%, c = 60.65%, d = 72.05%) as compared to CS (a).

**Figure 5 polymers-14-01707-f005:**
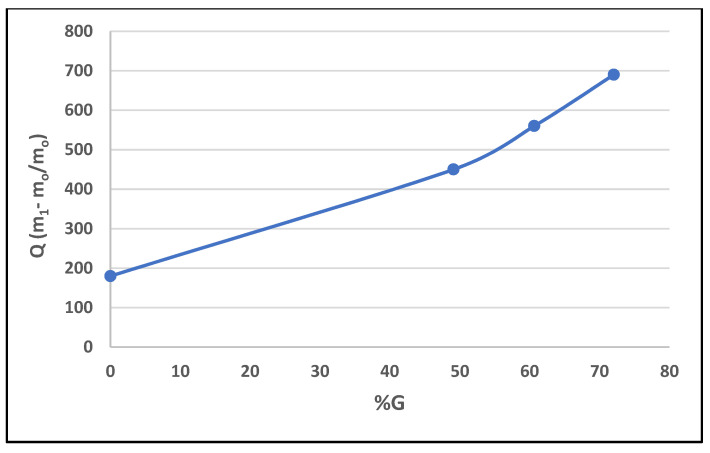
Water absorption capacity as a function of the percentage of graft.

**Figure 6 polymers-14-01707-f006:**
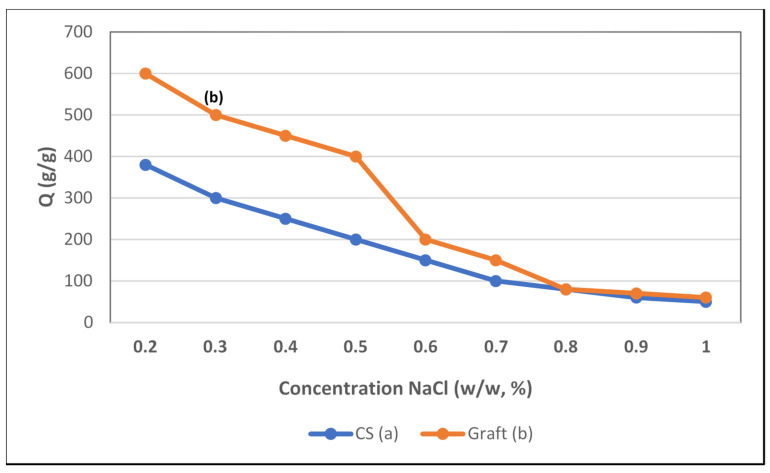
Water absorption capacity in different concentration salt solutions of superabsorbent (b) (%G = 60.65), as compared to CS (a).

**Figure 7 polymers-14-01707-f007:**
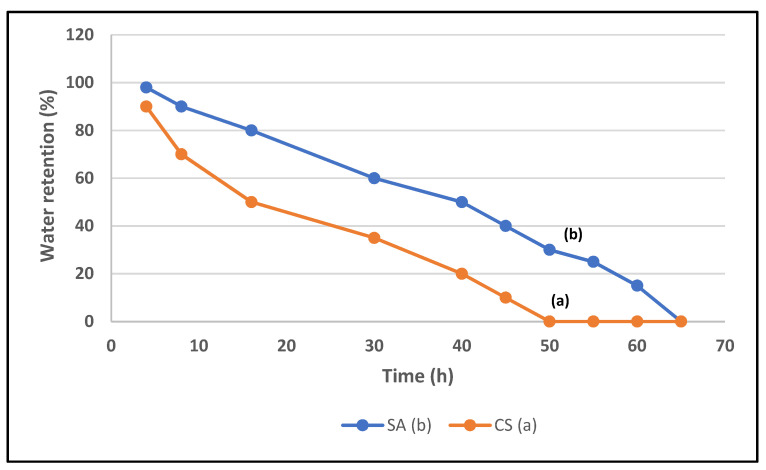
Water retention of SA (%G = 60.65) (b) as compared to that of CS (a).

**Table 1 polymers-14-01707-t001:** Water solubility of various synthesized (DAA-IA) copolymers.

Sample No.	In-Feed Composition (M)	Found Composition	Water Solubility
	(DAA, IA)	(DAA, IA)	
1	(0.8, 0.2)	(0.95, 0.05)	Insoluble
2	(0.6, 0.4)	(0.85, 0.15)	Insoluble
3	(0.5, 0.5)	(0.8, 0.2)	soluble
4	(0.4, 0.9)	(0.7, 0.3)	Soluble

**Table 2 polymers-14-01707-t002:** ^13^C-NMR spectral data of copolymerization of IA and DAA.

Structure	Carbon Atoms	^13^C-NMR (δ ppm)
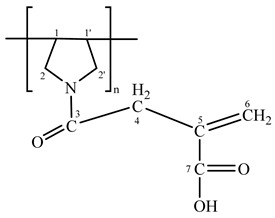	C1, C1′	40
	C2, C2′	55
	C3	173
	C4	31
	C5	133
	C6	123
	C7	177

**Table 3 polymers-14-01707-t003:** Effect of time on the graft percentage.

Grafting Time (h)	Grafting (%) (CS = 2 g, [DAA] = 0.4 M, [IA] = 0.15 M, [I] = 0.02 M, T = 60 °C)
0	0
1	13.7
2	25.85
3	38.85
4	38.75
5	38.5
6	38.45

**Table 4 polymers-14-01707-t004:** Effect of temperature on the graft percentage.

Temperature (°C)	Grafting (%) (CS = 2 g, [DAA] = 0.4 M, [IA] = 0.15 M, [I] = 0.02 M, Grafting Time = 3 h)
25	0
30	33.85
40	72.05
50	51.45
60	38.75

**Table 5 polymers-14-01707-t005:** Effect of initiator concentration on the percentage of graft.

Concentration of Initiators (M)	Grafting (%) (CS = 2 g, [DAA] = 0.4 M, [IA] = 0.15 M, Grafting time = 3 h, T = 40 °C)
0.015	60.65
0.02	72.05
0.025	49.65

## Data Availability

Not applicable.

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
