# Peer review of "Free Radical Copolymerization of Diallylamine and Itaconic Acid for the Synthesis of Chitosan Base Superabsorbent"

_polymers, 2022, doi:10.3390/polym14091707_

Round 1

Reviewer 1 Report

Dear Authors,
All my comments on the substantive content can be found in the attached file, however, in my opinion, for higher quality of your work it should be complemented with a series of structural and physicochemical tests.

Best regards,

Reviewer 2 Report

Reviewers' comments

In this manuscript: “A novel superabsorbent copolymer was successfully synthesized by the free radical polymerization of chitosan with dialylamine and itaconic acid comonomers. The reaction conditions were optimized.” The manuscript is innovative to a certain extent, but it also has the following shortcomings:

  1. In terms of manuscript, figures and formulas, the author made many mistakes in details in the writing process, the manuscript should be carefully checked and revised. For example:

    (1) On page 2, lines 74 and 75, this is the first formula in the manuscript, but the authors did not sort it and suggest reordering the formulas in this paper. And there are irregularities in the writing of relevant physical quantities, “W0 = original weight of chitosan w = weight of the graft copolymer”

   W0 is different from the writing in the formula. Formulas (1) and (2) also have related writing irregularities. It is recommended to carefully modify these physical quantities.

   (2) On page 5, line 135, 141, 149, On page 6, line 157, "CS = 2g, time= 3h", the normative writing between various related numbers and physical quantities.

   (3) In manuscripts, the writing of punctuation marks in language expressions is incomplete, and there are many problems. “On page 2, lines 74, 78” “On page 3, lines 94, 102, 110” “On page 6, lines 160” “On page 8, lines 193”

   (4)  Figure 3, what substance each line represents, it should be modified.

         Figure 6, the abscissa in the figure is not standardized and suggested to be revised.

         Figure 4 and Figure 5, the ordinate on the picture is incomplete and should be modified.

  1. In the manuscript, only by 13C-NMR spectrum to characterize and confirm the structure of the copolymer, and the superabsorbent polymer, it is not sufficient to demonstrate the successful progress of the polymerization reaction. In order to fully prove the correctness of the structure, it is recommended to supplement the infrared spectra of these polymers.

     In addition, what solvent is used in the manuscript to dissolve these polymers and characterize them by 13C-NMR spectrum.

  1. In order to further improve the aesthetics of the manuscript, it is recommended to use relatively standard software for drawing all the figures in the manuscript, such as origin software. And the fonts in the figure should be unified, and there should not be multiple fonts in the same manuscript at the same time.
  2. To clearly illustrate the microscopic morphology of these polymers, the SEM, AFM, or TEM spectra of these polymers should be given in the manuscript.

     5. When discussing the factors affecting the water absorption capacity of the superabsorbent polymers, it is recommended to supplement the initiator dosage.

Round 2

Reviewer 2 Report

Reviewers' comments

The standardization of the manuscript can make readers more clearly understand the content of the article. According to the comments made by the reviewer for the first time, the author has made some revisions to the manuscript, but some places have not been modified. Therefore, in order to improve the quality of the manuscript, it is suggested that the author of the manuscript further modify the manuscript.

This author should further modify the following issues:

(1) All formulas in the manuscript should be numbered uniformly. Two different serial numbers should not appear in the same manuscript.

(2) Authors are advised to carefully check whether there are spaces between all numbers and physical units in the manuscript. Consistency should be maintained in the manuscript.

(3) Figure 5, the name of ordinate should be carefully checked again and has not been modified.

For example:   Ordinate: Q (m1-m0/m0  v
